# Assessing the Level of Awareness of COVID-19 and Prevalence of General Anxiety Disorder among the Hail Community, Kingdom of Saudi Arabia

**DOI:** 10.3390/ijerph18137035

**Published:** 2021-06-30

**Authors:** Bandar Alsaif, Najm Eldinn Elsser Elhassan, Ramaiah Itumalla, Kamal Elbassir Ali, Mohamed Ali Alzain

**Affiliations:** 1Department of Public Health, College of Public Health and Health Informatics, University of Hail, Hail 55476, Saudi Arabia; b.alsaif@uoh.edu.sa (B.A.); n.hamedelneel@uoh.edu.sa (N.E.E.E.); K.ali@uoh.edu.sa (K.E.A.); 2Department of Health Management, College of Public Health and Health Informatics, University of Hail, Hail 55476, Saudi Arabia; r.itumalla@uoh.edu.sa

**Keywords:** attitude, COVID-19, general anxiety disorder, Hail, knowledge, practice

## Abstract

Background: The COVID-19 pandemic has caused a major public health problem around the world. Therefore, the aim of the study was to assess the awareness and prevalence of General Anxiety Disorder (GAD) with regard to COVID-19 among the Hail community, in the Kingdom of Saudi Arabia, in order to help health authorities to effectively control the pandemic. Methods: A cross-sectional online survey was completed by 412 participants living in Hail, Saudi Arabia. The questionnaire assessed demographic characteristics, knowledge, attitudes, and practices for the prevention of COVID-19, as well as psychological feelings in terms of GAD as an impact of the COVID-19 pandemic. Results: The study found that most of the respondents demonstrated good knowledge, attitudes, and practice for COVID-19 prevention. The elderly and employed demonstrated significant positive attitudes and practices (*p* < 0.05). Participants with a positive attitude were almost two and a half times (OR = 2.4; 95% CI: 1.54–3.99) more likely to have good practices. Additionally, the rural respondents were less likely (OR = 0.45; 95% CI: 0.21–0.96) to have a positive attitude. Married participants were more than one and a half (OR = 1.60; 95% CI: 1.04–2.44) times more likely to have a positive attitude. The prevalence of GAD was 21.8% and was significantly increased among participants with inadequate knowledge (OR = 2.01; 95% CI: 1.25–3.22), females (OR = 1.92; 95% CI: 1.19–3.09), individuals with chronic diseases (OR = 1.71; 95% CI: 1.02–2.86), and non-Saudi participants (OR = 2.44; 95% CI: 1.02–5.85). Conclusions: Ensuring a sufficient combination of relatively good levels of knowledge, positive attitudes, and desired practices serves as a good approach to preventing the spread of COVID-19. However, the increased prevalence of GAD requires the attention of policymakers. Therefore, a great emphasis should be placed on health awareness campaigns, with a focus on misconceptions and the provision of counseling.

## 1. Introduction

The outbreak of the coronavirus disease (COVID-19) pandemic has posed a major public health risk across the globe, including in the Kingdom of Saudi Arabia (KSA). COVID-19 has spread rapidly all around the globe. As per the World Health Organization’s (WHO) latest updates, there have been 112,209,815 confirmed cases and 2,490,776 confirmed deaths due to COVID-19, reported across 223 countries, areas or territories [1]. Though the KSA has made extensive efforts to stop the entry of COVID-19 into the country, it joined the list of countries affected by COVID-19 on 2 March 2020 [2]. Subsequently, COVID-19 cases have been increasing in the country and according to the Saudi Ministry of Health (MOH), there have been 376,377 confirmed cases along with 6480 death (as of 26 February 2021) in the country [3]. Out of the total number of COVID-19 cases, 367,323 individuals recovered and at present there are 2574 active cases in the country [1].

COVID-19 is spread by person-to-person transmission and by contact with contaminated surfaces or objects. Primarily, the virus is spread between people who are in close contact with another infected individual and through the respiratory droplets produced when an infected person coughs or sneezes. Transmission is also possible as the result of an individual touching a surface or object that is contaminated with the virus and then touching their mouth, nose, or—most likely—eyes [4,5]. According to the WHO, physical distancing, wearing a mask, ensuring that rooms are well ventilated, avoiding crowds, hands washing, and coughing into a bent elbow or tissue are some of the simple precautions one can take to protect themselves and others from COVID-19 [6].

The government of Saudi Arabia has been routinely taking all possible measures to control COVID-19; however, people’s support plays a vital role in the ongoing process [2]. The study of public awareness and attitudes towards the disease will help health authorities to construct effective policies to manage COVID-19 [7]. The knowledge, attitudes, and practices (KAP) studies suggest that adherence to mandated control measures are important in preventing the spread of the virus, but that this is also dependent on the population’s overall KAP [8,9,10]. Besides representing one of the greatest risks to physical health, COVID-19 presents a danger to the mental health of the population [11]. The majority of studies on mental health issues during COVID-19 are derived from China [12,13], with very few studies of Western populations having been conducted [13,14,15]. In this context, there is a pressing need to study the impact of the COVID-19 pandemic on mental health in the Gulf region, in particular, in the Saudi Arabian population. Thus, the present study aims to fill this research gap.

The study of people’s awareness of COVID-19 and General Anxiety Disorder (GAD) may serve the dual purpose of helping health authorities to effectively control the pandemic and understand the GAD status among communities. This will lead to the ability to provide the required services to people with GAD. Hence, the present study aimed to assess the level of awareness towards COVID-19 and to investigate the psychological impact of the COVID-19 pandemic among the Hail community in the Kingdom of Saudi Arabia.

## 2. Materials and Methods

### 2.1. Study Design

The present study is an analytical cross-sectional community-based study, conducted as an online survey in various governorates of the Hail community from 1 June 2020 to 27 June 2020. This was the preferred method due to the prevalence of COVID-19 in the country, resulting in the inability to conduct an interviewer-administered questionnaire.

### 2.2. Study Area and Population

Hail is a region of Saudi Arabia, located in the northwest of the country. It has an area of 120,000 km^2^ and a population of 716,021, as of the 2018 census [15]. It is subdivided into 8 governorates [15]. The study involved populations aged ≥18 years living in Hail.

### 2.3. Sampling Techniques

The minimum sample size was (398) calculated using the following formula: (1)$$n=\frac{N}{1+N{\left(e\right)}^{2}}$$
*n* = sample size; *N* = total population; *e*^2^ = level of precision = 0.05. A total of 412 completed questionnaires were received and included in the study using a convenient sampling method.

### 2.4. Data Collection

The survey was designed in webpage format using Google Form. The link was distributed via social media (Twitter, Instagram, Facebook and WhatsApp, etc.) in order to achieve wider circulation to reach all participants in the Hail region during the COVID-19 pandemic. The questionnaire contained three sections: (1) demographic characteristics (gender, age, occupation, marital status, educational level, and residence); (2) a section on the level of knowledge, attitudes, and practice against the prevention of COVID-19. Knowledge was defined as the awareness of the population in relation to preventing the spread of COVID-19 [16]. The knowledge section had 10 questions; each question was answered “yes”, “no” or “I don’t know”. “Yes” scored 1, while “no” and “I don’t know” scored 0. Attitudes were defined as the way the population thinks and behaves in relation to preventing the spread of COVID-19 [16]. This was measured by seven questions using a three-point Likert’s scale “agree” (1 point); either “undecided” or “disagree” (0 points) for active questions. Practice was defined as the habits of the population related to preventing the spread of COVID-19 [16]. The practice section had seven items, and each item was answered “always” (2 points), “sometimes” (1 point) or “never” (0 points).

To confirm the quality of data collection, experts in the public health, health promotion, and epidemiology fields reviewed the questionnaire and conducted pretests using Cronbach’s alpha test to measure the reliability (0.79). The third section focused on psychological feelings in terms of GAD, and as the impacts of the COVID-19 pandemic.

### 2.5. Data Management and Analysis

The data were analyzed using SPSS version 25. Frequencies, percentages, and chi-square tests were used to identify the significant demographic variables associated with KAP toward COVID-19 prevention. The total scores for knowledge were computed, ranging between (0–10), with the mean score of 07.44 ± 1.02 being used to classify the knowledge. Participants with ≥8 (with 74.4% correct answers) were considered to have a good level of knowledge. The mean score for attitudes was 5.25 ± 1.37, ranging from 0–7. Participants with scores ≥6 (with 75.00% correct answers) were considered to have a positive attitude. While the mean score for practice was 12.75 ± 1.88, ranging from 0–14. Participants with scores of ≥13 (with 91.01% correct answers) were considered to have good practices. A binary logistic regression test was used to detect significant demographic variables associated with attitudes and practices. All results are presented in terms of coefficients (B) and odds ratio (OR) with confidence intervals (CI). All tests were two-sided and *p* < 0.05 was considered significant.

### 2.6. Ethical Approval

All the procedures performed in this study abided by ethical standards. Ethical approval was obtained from the Research Ethics Committee at the University of Hail (UOH), KSA (Nr. 46123/5/41). Informed consent was included in the survey’s cover page (purpose, procedures, guarantee of anonymity, and confidentiality). Those who agreed to participate in the study were instructed to click the link and complete the questionnaire.

## 3. Results

Out of the 412 respondents who completed the online survey; (41.5%) were in the age group 26–49 years. A total of 51.5% of respondents were male, and more than one-third (36.4%) were students, while 33.3% were employed. More than half (53.9%) of the study population were single, while the majority (93%) lived in Hail city. Moreover, approximately a quarter (24%) of respondents had chronic diseases, as shown in (Table 1).

The study found that 59% of respondents had good knowledge of COVID-19 prevention (total score ≥ 82.7%). Knowledge increased significantly with age, while housewives and the employed had higher percentages of good knowledge levels than other occupational groups. The study found that the married and divorced groups had significantly higher levels of good knowledge compared to the single group (*p*-value < 0.05).

The study found that 69.2% of the respondents had positive attitudes in terms of preventing the spread of COVID-19 (total score ≥ 87.5%). Positive attitudes increased significantly with age, while individuals in certain occupations (free businesses, retired, and employed) demonstrated more positive attitudes compared to those in other occupations. The study also found that married individuals have significantly more positive attitudes compared to single individuals (*p* < 0.05) (Table 1).

Furthermore, our study reveals that 68% of respondents demonstrated good practices towards preventing the spread of COVID-19 (total score ≥ 91%). The rate of good practices increased significantly with age, while individuals of certain occupations (free business, retired, and housewife) were more likely to demonstrate good practices compared to other occupations. There was no significant difference in COVID-19 practices between genders, residence, marital status, and chronic disease status (*p*-value < 0.05), as shown in Table 1.

The regression analysis found that there was no significant association between good knowledge with attitudes and practices towards preventing the spread of COVID-19. While the elderly (>50 years) were more than two (OR = 2.18; 95% CI: 1.17–4.08) times as likely to have a positive attitude compared to the younger age groups (18–25 years). Moreover, the employed were almost two (OR = 1.72; 95% CI: 1.02–2.88) times as likely have a positive attitude, while the unemployed were less likely (OR = 0.16; 95% CI: 0.05–0.46) to have a positive attitude compared to students. Rural respondents were less likely (OR = 0.45; 95% CI: 0.21–0.96) to have a positive attitude than those living in a city. Additionally, married participants were more than one and a half (OR = 1.60; 95% CI: 1.04–2.44) times more likely to have a positive attitude than those who were single.

Moreover, participants with a positive attitude were almost two and a half (OR = 2.4; 95% CI: 1.54–3.99) times more likely to demonstrate good practices than those with a negative attitude. Additionally, good practice correlated significantly with increased age; 26–49 and >50-year age groups (OR = 1.67; 95% CI: 1.05–2.63) and (OR = 2.52; 95% CI: 1.37–4.65) were more likely to have good practices compared to younger groups (18–25 years), respectively. Additionally, certain groups such as those who were employed or retired were almost two and three times as likely (OR = 1.80; 95% CI: 1.09–2.96) and (OR = 2.67; 95% CI: 1.24–5.74), respectively, to have good practices compared to students. In addition, married participants were more than one and a half (OR = 1.64; 95% CI: 1.07–2.50) times as likely to have good practices compared to those who were single, as shown in Table 2.

The study found that the prevalence of GAD among respondents was 21.8%, as shown in Figure 1.

Table 3 reveals that participates with inadequate knowledge were twice (OR = 2.01; 95% CI: 1.25–3.22) more likely to have GAD than those with good knowledge. Moreover, older, and married participants were less likely (OR = 0.36; 95% CI: 0.17–0.76 and OR = 0.46; 95% CI: 1.07–2.50, respectively) to have GAD compared to those who were younger and single. While females were almost twice (OR = 1.92; 95% CI: 1.19–3.09) more likely to have GAD than males. Additionally, participants with chronic diseases were almost twice (OR = 1.71; 95% CI: 1.02–2.86) more likely to have GAD than those free from chronic diseases. Likewise, non-Saudi participants were two and a half (OR = 2.44; 95% CI: 1.02–5.85) times more likely to have GAD than Saudi participants.

## 4. Discussion

Mitigating the impacts of COVID-19 on people’s mental health disorders and the awareness of the population have significant effect on the ability to combat COVID-19 [17]. Therefore, there is a need to assess the general level of awareness and prevalence of GAD in response to COVID-19 in the Hail region, Saudi Arabia.

The knowledge of respondents was relatively good, similar to the findings of most studies conducted in several countries [18,19]. This finding could be due to ongoing awareness campaigns conducted by the Ministry of Health (MOH) on combating the COVID-19 pandemic. The MOH has published awareness guidelines in numerous languages [18]. Subsequently, our findings similar to a previous study conducted in Riyadh, the Capital of Saudi Arabia [4].

Our study highlighted that elderly had a good level of knowledge and attitudes, which subsequently led to increased compliance with the preventive measures of COVID-19. These results align with previous studies that confirmed that the elderly are more likely to have a good level of knowledge and practice concerning COVID-19 [4,20]. This finding could be due to that elderly people have the cognitive ability to assess available COVID-19 information and use it to adopt positive beliefs and practices toward their health [21]; conversely, younger people commonly display risk-taking behavior toward their health [22]. Furthermore, the elderly are responsible for their families and are caregivers for children, so they show better attitudes and compliance toward preventive measures. In contrast to those studies, a study conducted in Egypt found that elders have lower levels of knowledge compared to younger generations [5]. The observed contradiction could be due to differences in socioeconomic status between the two countries.

Our study found a significant association between a good level of knowledge and being housewives, employed, and divorced; furthermore, most of these groups in our study were in the >35 years age group, which appeared to demonstrate a good level of good knowledge. 

The positive attitude of participants was generally good in terms of preventing the spread of COVID-19. This finding is consistent with studies conducted in China, Sudan, Riyadh, and Saudi Arabia [3,4]. However, respondents’ attitude scores were higher than in the study conducted in Bangladesh [21]. In our study, positive attitude scores were significantly increased among employed and married; these results are in line with several studies conducted in Saudi Arabia [3,4,18]. Moreover, the majority of married and employed individuals in our study were elderly, who are known to have increased positive attitudes towards the prevention of COVID-19. Based on our finding, the participants from the rural area was found to have a lower level of attitude towards COVID-19. This finding might be due to limited access to the internet to gain health information to update themselves compared to the participant from urban areas. The internet (online newspapers and social networks) is one of the principal sources of COVID-19 information during the pandemic [23].

Overall, COVID-19 prevention practices were relatively good, which might be due to the Saudi government-imposed protective measures, such as social distance and wearing masks in public places, aimed at combatting COVID-19. Simultaneously, the percentage of good practice increased significantly with good attitudes, married, and employed. The majority of these groups are mature, which positively increases protective beliefs and behaviors towards COVID-19 prevention.

The findings of this study indicate that the prevalence of (GAD) among respondents was 21.8%, which is less than the overall prevalence of GAD globally [24,25]. Moreover, it was lower than reported in the earlier pandemic period in China (28.8) [26]. We also noticed that lower knowledge scores were associated with increased GAD prevalence compared to people with good knowledge scores [27]. A study conducted in China confirmed this finding, indicating that those knowledgeable people are associated with a lower psychological impact of the pandemic [26]. Furthermore, younger people of the Hail community were more likely to have GAD; similar results have been reported in the previous study from China [24]. This finding may be due to a lack of maturity in understanding the COVID-19 situation demonstrated by younger generations. The study also found that single people were more likely to have GAD compared to married and divorced, which may be due to loneliness and a lack of support during the pandemic. Furthermore, females were at a higher risk of being affected by GAD, and this could be due to the fact that females often take care of the entire family in Saudi Arabia—including looking after the children, cooking, etc.—while at home during the lockdown and the COVID-19 pandemic. A similar result was found in a study conducted in multinational middle-income countries in Asia found females, younger < 30 years, and single and separated status are more susceptible to develop psychological distress [28].

One of the important findings of this study is that people with chronic diseases were likely to have GAD; our previous study confirmed this finding, indicating that people with chronic diseases have a lower quality of life and a higher level of psychological distress during the pandemic [29]. Moreover, most people with chronic disease avoid medical visits during the COVID-19 pandemic, which leads to increased level of anxiety [30,31]. Therefore, patients with chronic diseases must continue to receive care, despite the pandemic [32]. Another vital finding of the study shows that the non-Saudi population of the Hail community were more likely to have GAD as compared to Saudi nationals. This could be due to factors such as the inability to go back to their country of origin due to the suspension of international flights, lack of work, and in some cases, due to an expired Visa. The lack of family support in non-Saudi populations during the pandemic might have resulted in the higher risk of developing GAD.

## 5. Conclusions

In general, the study found a relatively good level of knowledge, positive attitudes, and good practices related to preventing the spread of COVID-19. Therefore, an emphasis should be placed on health awareness campaigns focusing on those (students, younger, and those living in rural areas) engaged in risk-taking practices and removing misconceptions related to social distancing, hand shaking, and mask wearing. The prevalence of Generalized Anxiety Disorder (GAD) among respondents of the Hail community requires the attention of healthcare policy-makers, especially among the younger aged, females, people with chronic diseases, and foreigners.

## Figures and Tables

**Figure 1 ijerph-18-07035-f001:**
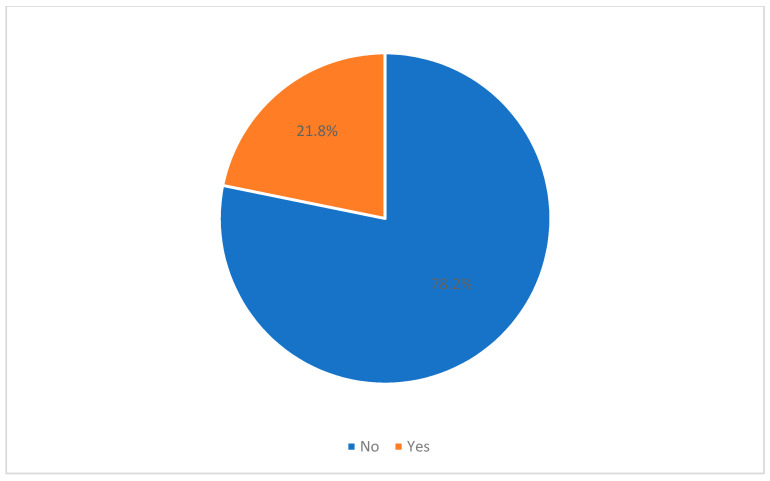
Prevalence of General Anxiety Disorder (GAD) among Hail population during COVID-19 pandemic.

**Table 1 ijerph-18-07035-t001:** Demographic characteristics and knowledge, attitudes, and practice classifications.

Variables	Total	Knowledge		Attitude		Practice	
	*n* (%)	Good *n* (%)	Bad *n* (%)	Positive *n* (%)	Negative *n* (%)	Good *n* (%)	Bad *n* (%)
**Overall**	412 (100)	243 (59)	169 (41)	285 (69.2)	127 (30.8)	280 (68.0)	132 (32.0)
**Age group**							
18–25 years	157 (38.1)	78 (49.7)	79 (50.3) **	101 (64.3)	56 (35.7) *	(59.2) 93	64 (40.8) **
26–49 years	171 (41.5)	109 (63.7)	62 (36.3)	117 (68.4)	54 (31.6)	121 (70.8)	50 (29.2)
>50 years	84 (20.4)	56 (66.7)	28 (33.3)	67 (79.8)	17 (20.2)	66 (78.8)	18 (21.4)
**Gender**							
Male	212 (51.5)	129 (60.8)	83 (39.2) ^ns^	(70.8) 150	62 (29.2) ^ns^	137 (64.6)	75 (35.4) ^ns^
Female	200 (48.5)	114 (57.0)	86 (43.0)	135 (67.5)	65 (32.5)	143 (71.5)	57 (28.5)
**Occupation**							
Student	150 (36.4)	73 (48.7)	77 (51.3) *	97 (64.7)	53 (35.3) ***	90 (60.0)	60 (40.0) **
Employee	137 (33.3)	91 (66.4)	46 (33.6)	104 (75.9)	33 (24.1)	100 (73.0)	37 (27.0)
Free business	10 (2.4)	6 (60.0)	4 (40.0)	8 (80.0)	2 (20.0)	8 (80.0)	2 (20.0)
Housewife	43 (10.4)	31 (72.1)	(27.9) 12	32 (74.4)	11 (25.6)	32 (74.4)	11 (25.6)
Unemployed	22 (5.3)	14 (63.6)	8 (36.4)	5 (22.7)	17 (77.3)	10 (45.5)	12 (54.5)
Retired	50 (12.1)	(56.0) 28	22 (44)	39 (78.0)	11 (22.0)	40 (80.0)	10 (20.0)
**Marital status**							
Married	180 (43.7)	90 (50.0)	90 (50.0) **	113 (62.8)	67 (37.2) **	112 (62.2)	68 (37.8) ^ns^
Single	222 (53.9)	144 (64.9)	78 (35.1)	162 (73.0)	(27.0) 60	162 (73.0)	60 (27.0)
Divorce	10 (2.4)	9 (90.0)	1 (10.0)	10 (100)	0	6 (60.0)	4 (40.0)
**Residence**							
City	383 (93.0)	229 (59.8)	154 (40.2) ^ns^	270 (70.5)	113 (29.5) *	264 (68.9)	119 (31.1) ^ns^
Village	29 (7.0)	14 (48.3)	15 (51.7)	15 (51.7)	14 (48.3)	16 (55.2)	13 (44.8)
**Chronic disease**							
Yes	99 (24.0)	66 (66.7)	33 (33.3) ^ns^	71 (71.7)	28 (28.3) ^ns^	205 (65.5)	108 (34.5) ^ns^
No	313 (76.0)	177 (56.5)	136 (43.5)	214 (68.4)	99 (31.6)	75 (75.8)	24 (24.2)

Level of statistical significance: ^ns^ >0.05; * <0.05; ** <0.01; *** <0.001.

**Table 2 ijerph-18-07035-t002:** Binary logistic regression to predict participants with positive attitudes and good practices in the Hail region.

Variables	Attitude	Practice
	OR (95% CI)	OR (95% CI)
**Knowledge**		
Bad	Reference	Reference
Good	1.45 (0.95–2.21) ^ns^	1.50 (0.98–2.27) ^ns^
**Attitude**		
Negative	Reference	Reference
Positive	-	2.40 (1.54–3.99) ***
**Age Group**		
18–25 years	Reference	Reference
26–49 years	1.20 (0.76–1.90) ^ns^	1.67 (1.05–2.63) *
>50 yeas	2.18 (1.17–4.08) *	2.52 (1.37–4.65) **
**Occupation**		
Student	Reference	Reference
Employee	1.72 (1.02–2.88) *	1.80 (1.09–2.96) *
Free Business	2.18 (0.45–10.6) ^ns^	2.66 (0.55–12.99) ^ns^
Housewife	1.59 (0.74–3.41) ^ns^	1.94 (0.91–4.14) ^ns^
Unemployed	0.16 (0.05–0.46) **	0.56 (0.22–1.37) ^ns^
Retired	1.94 (0.91–2.10) ^ns^	2.67 (1.24–5.74) *
**Residence**		
City	Reference	Reference
Village	0.45 (0.21–0.96) *	0.55 (0.26–1.9) ^ns^
**Marital Status**		
Single	Reference	Reference
Married	1.60 (1.04–2.44) *	1.64 (1.07–2.50) *
Divorce	0.99 (0.10–2.21) ^ns^	0.91 (0.25–3.34) ^ns^

Degree of statistical significance: ^ns^ >0.05; * <0.05; ** <0.01 *** <0.001; OR: odds ratio; CI: confidence interval.

**Table 3 ijerph-18-07035-t003:** Binary logistic regression to predict participants with anxiety disorder in the Hail region.

Variables	Anxiety Disorder		
	Yes *n* (%)	No *n* (%)	OR (95% CI)
**Total**	90 (21.8)	322 (78.2)	-
**Knowledge**			
Good	41 (16.9)	202 (83.1)	Reference
Bad	49 (29.0)	120 (71.0)	2.01 (1.25–3.22) **
**Age Group**			
18–25 years	43 (27.4)	114 (72.6)	Reference
26–49 years	37 (21.6)	134 (78.4)	0.73 (0.44–1.21) ^ns^
>50 yeas	10 (11.9)	74 (88.1)	0.36 (0.17–0.76) **
**Gender**			
Male	35 (16.5)	177 (83.5)	Reference
Female	55 (27.5)	145 (72.5)	1.92 (1.19–3.09) **
**Marital Status**			
Single	53 (29.4)	127 (70.6)	Reference
Married	36 (16.2)	186 (83.8)	0.46 (1.07–2.50) **
Divorce	1 (10.0)	9 (90.0)	0.26 (0.03–2.15) ^ns^
**Chronic disease**			
No	61 (19.5)	252 (80.5)	Reference
Yes	29 (29.3)	70 (70.7)	1.71 (1.02–2.86) *
**Nationality**			
Saudi	81 (20.8)	308 (79.2)	Reference
Non-Saudi	9 (39.1)	14 (60.9)	2.44 (1.02–5.85) *

Degree of statistical significance: ^ns^ >0.05; * <0.05; ** <0.01; OR: odds ratio; CI: confidence interval.

## Data Availability

The data are not visibly available due to ethical agreements at the time.

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
