# Peer review of "Assessing the Level of Awareness of COVID-19 and Prevalence of General Anxiety Disorder among the Hail Community, Kingdom of Saudi Arabia"

_ijerph, 2021, doi:10.3390/ijerph18137035_

Round 1

Reviewer 1 Report

The current article by Bandar Alsaif and al., reviews the current literature on the correlation between COVID-19 with prevalence of General Anxiety Disorder (GAD) among the Hail community, in the Kingdom of 14 Saudi Arabia. The title of the paper is in line with the body of the manuscript. The topic is current and timely and very important for the world's scientific community in the current period of the global COVID-19 pandemic. The authors have written a clear and detailed review and the material is well presented. The references used are suitable and it is new and updated material, I believe that the article can be accept pending other revision and I only have one observation to make:

  1. The percentage of GAD (21,2%) between line (186) and percentage in fig. 2 (21,8)% is different

Author Response

Point 1: English language and style are fine/ minor spell check required

Response1: Thank you for the appreciation. The manuscript was edited for proper English language, grammar, punctuation, spelling, and overall style. We hope it now matches the journal requirements.

Point 2: the percentage of GAD (21.2%) and in figure2 (21.8%)

Response 2: Thank you for pointing this out; we corrected the differences in the results section. [page number 5, line 189]

Reviewer 2 Report

Dear authors,

The revised manuscript is interesting in the context of COVID-19 as it takes into account another population in order to confirm mental health effects, including awareness. It is well written and formally well conducted. However, there are several areas for improvement.

Firstly, I have not seen the dates on which the study was done. This is important since the impact of covid depends on uncertainty, and at the beginning there was a lot of uncertainty and lack of knowledge, and now there is much more knowledge. Also, if the study was done when vaccination started, it is possible that the impact on mental health is not comparable to that found in other studies when there was no vaccination or more uncertainty. In this sense, once the date of the study has been indicated, this aspect should be discussed in depth.

On the other hand, the discussion, in my opinion, is poor. It is limited to repeating the results and comparing them with the results of other studies, but only in terms of prevalence. However, I believe that the discussion could be improved by analysing the results in more depth, especially by analysing the role of age, education, marital status, etc. In fact, I think this is the most interesting aspect, in terms of proposing prevention and information programmes, etc. I therefore believe that the implications of these results should be discussed further.

Finally, as a minor point, there is the issue of income, the results of which are not clear. I think that these data (and the discussion about it)should be removed as they are not worked on any more, except to say that there has been a population that has lost more or less due to covid.

Author Response

Point 1: Methods section can be improved

Response 1: Thank you for pointing this out; we review the method for further improvement.

Point 2: conclusion can be improved

Response 2: Thank you for the comment. Conclusion section has been revised by adding one sentence to summarize the overall findings.

Point 3: I have not seen the date on which the study was done

Response 3: You have raised an important point. We added the survey date on the methods section [page number 2, line 79-80]

Point 4: discussion could be improved in more depth, especially the role of age, education, marital status, etc.

Response 4: Thank you very much for this valuable comment and for allowing us to respond to it. We have revised the ‘Discussion’ section by adding and deleting some sentences and paragraphs to come out more clearly, especially the role of demographic variables in proposing prevention programs.

Point 5: There is the issue of income, the results of which are not clear. I think the data should be removed.

Response 5: Thank you for this excellent suggestion; we removed the results of income loss (figure1) and also from the text; as it is not discussed.

Round 2

Reviewer 2 Report

In my first revision I suggested to discuss deeply the results. In the new version of the manuscript, discussion section is the same (only minor changes). 

Author Response

Point 1: improvement of discussion by deeply discussing the results in more depth, especially the role of age, education, marital status, etc.

Response 1: Thank you very much for this valuable comment and for allowing us to respond to it. We have revised the ‘Discussion and conclusion’ sections by adding and deleting some sentences and paragraphs to come out more clearly, especially the role of demographic variables in proposing prevention programs, and improving the results more in-depth especially the role of age, education, marital status, etc.